# "Lichenoid Granulomatous Pattern" in a Case of Lupus Vulgaris

Chirag Desai [1,*] and Ismail Shaikh [2]

1 Divya Sparsh Skin & Hair Clinic, Dadar, Mumbai 400014, India
2 Aesthederm Skin, Hair & Laser Clinic, Colaba, Mumbai 400005, India; dermis1@gmail.com
* Correspondence: 83.chirag@gmail.com

**Abstract:** Lupus vulgaris is a one of the most common skin infections in the Indian subcontinent. Even today, it often creates a diagnostic dilemma for both clinicians and histopathologists. We describe a case of lupus vulgaris that showed lichenoid granulomatous inflammation in the dermis. This pattern is not uncommon, but is rarely described in the literature as newer modalities currently take precedence in diagnosis. Our aim is to make clinicians and dermatopathologists aware of this pattern of inflammation seen in this common infection.

**Keywords:** lupus vulgaris; lichenoid granulomatous dermatitis; pattern of inflammation

## 1. Introduction

Lupus vulgaris is the most common manifestation of cutaneous tuberculosis in the Indian subcontinent. It is a chronic, paucibacillary form of cutaneous tuberculosis, occurring in all age groups with a slight female preponderance. The mechanism of spread is by lymphatic, hematogenous or direct contiguous spread from an adjacent site. Complete healing usually does not occur without appropriate therapy, although partial regression is possible with healing by scarring. Histopathological evaluation of a biopsy specimen is by far the most important modality for diagnosis of this entity. We present a case of lupus vulgaris with an unusual pattern in the histopathology, which is not described in classical textbooks of dermatopathology but is quite common, especially in the Indian subcontinent.

## 2. Case Details

A 13-year-old boy presented with an asymptomatic, scaly lesion on the right elbow that had been present for four months. The lesion was slowly progressive. On examination, there was a well-defined erythematous scaly plaque with scarring at one end present over the right elbow. The surface of the lesion also showed a few crusts and a small ulcer at the border of the opposite end. The lesion was non-tender on palpation. No enlargement of the surrounding lymph nodes was noted. Systemic examination did not reveal any abnormality. Clinical differential diagnoses of lupus vulgaris and deep fungal infection were considered, and a punch biopsy specimen was sent for histopathology evaluation. The Mantoux test was positive (10 mm) in this patient (Figure 1).

Biopsy showed a dense lichenoid and loose tuberculoid granulomatous infiltrate comprising of epithelioid cells, histiocytes and occasional Langhan's giant cells, surrounded by lymphocytes and few plasma cells. The granuloma was seen abutting the overlying epidermis, which showed psoriasiform hyperplasia with mild to moderate spongiosis. A compact tuberculoid granuloma was also seen in the deep reticular dermis. Fibroplasia was also seen in the dermis (Figures 2–5). The PAS-stained sections were negative for fungal organisms. Zeil–Neelsen stained sections did not reveal any acid-fast bacillus. Based on these features, a diagnosis of lupus vulgaris was favoured and further confirmation by culture study and PCR was advised.

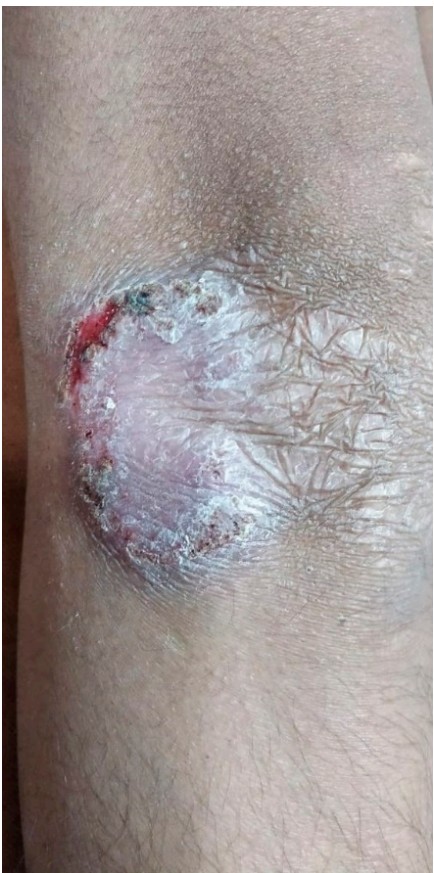

**Figure 1.** Well-defined scaly erythematous plaque with scarring at one end located on elbow.

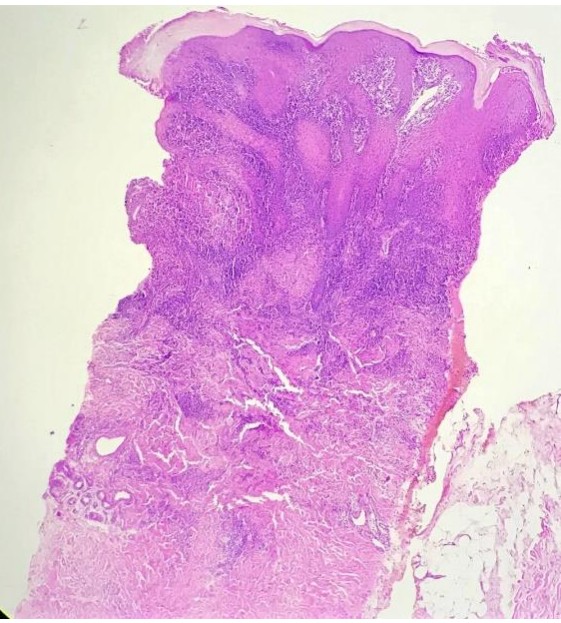

**Figure 2.** Lichenoid granulomatous dermatitis pattern with psoriasiform epidermal hyperplasia (H&E × 40).

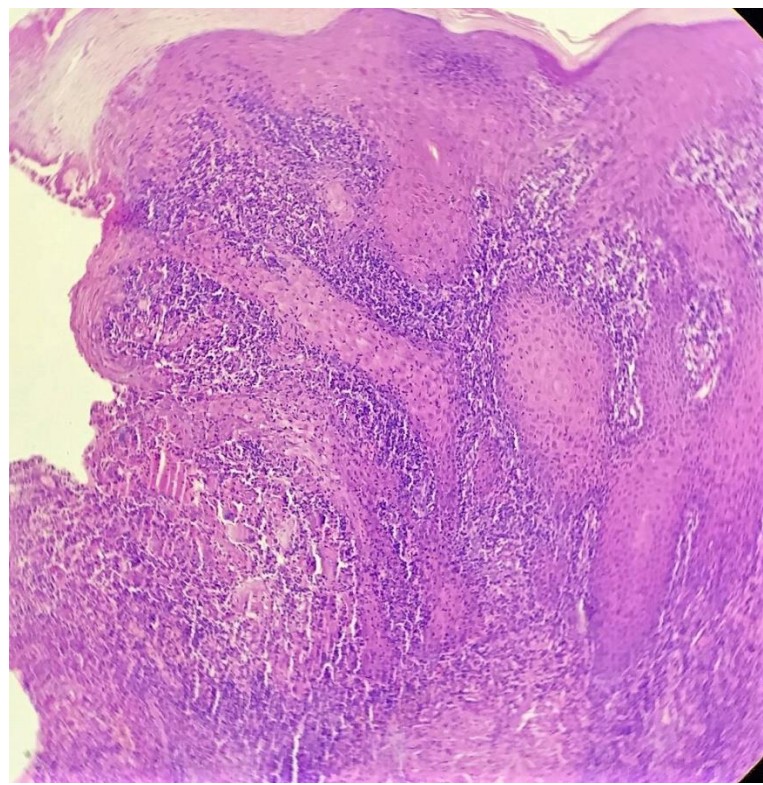

**Figure 3.** Loose tuberculoid granuloma abutting the overlying hyperplastic epidermis (H&E × 100).

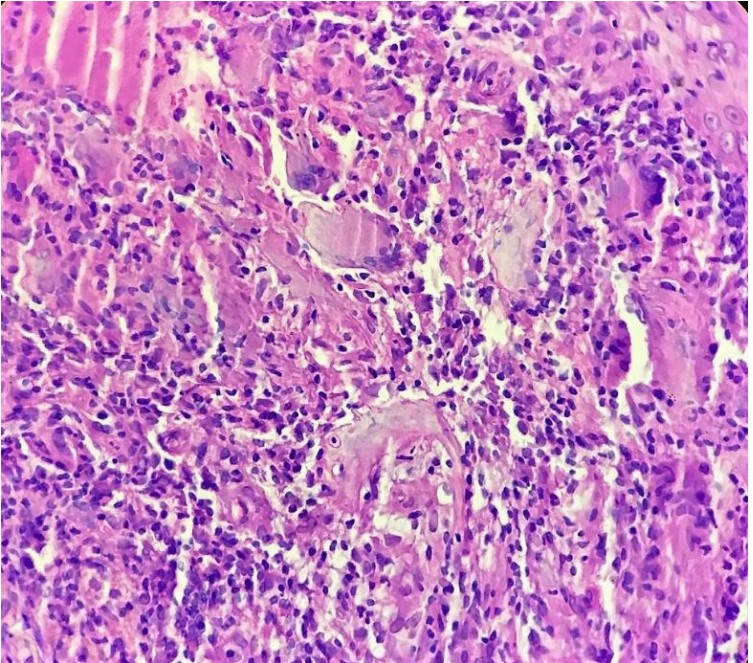

**Figure 4.** Tuberculoid granuloma in upper dermis comprised of epithelioid cells, giant cells and surrounded by lymphocytes (H&E × 400).

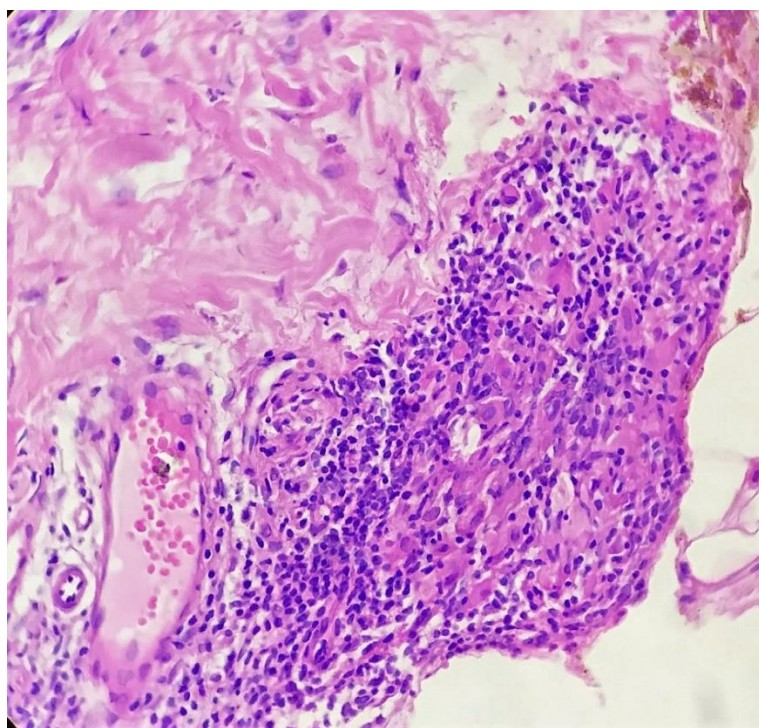

**Figure 5.** Compact tuberculoid granuloma in deep dermis (H&E × 400).

However, since the parents of the patient were not able to afford the further testing, it could not be done. Based on the clinical–pathological correlation, an empirical trial of anti-tubercular drugs was given, after which the patient showed response at four weeks so it was continued up to six months as per the protocol. The lesion was nearly regressed at six months.

### 3. Discussion

Cutaneous tuberculosis is one of the most common skin infections occurring in the Indian subcontinent. Tuberculoid granulomas with or without central caseation necrosis and with secondary epidermal changes are usually present in the histology in most cases of lupus vulgaris. It is difficult to find the organism in the biopsy specimen, even with the modified Zeihl–Neelsen's staining method, which causes a diagnostic dilemma for a histopathologist. A culture study and PCR is then usually advised for confirmation. However, even these modalities may have false negative outcomes and may be inaccessible in limited set ups; this poses a diagnostic dilemma for the clinician. Therefore, a four-to-six-week trial of anti-tubercular drugs in clinically suspicious cases with supportive histopathology is a generally acceptable method of "diagnosis", especially in resource-poor settings [1].

The lichenoid granulomatous dermatitis pattern (LGD) in the histopathology is seen in many diverse conditions such as drug eruption, lichenoid keratosis, tattoo reaction, post-herpetic dermatitis, scabies and post-scabetic dermatitis, pigmented purpuric dermatosis, lichen striatus, lichenoid sarcoidosis and secondary syphilis [2–4]. It has also been previously described in other mycobacterial infections such as tuberculoid leprosy [2], atypical mycobacterial infections due to Mycobacterium marinum [5] and Mycobacterium kansasii [5], and also in Mycobacterium haemophilum [6] infection in leukaemia or non-Hodgkin lymphoma.

The LGD pattern has also been described in lupus vulgaris in some previous studies [7], and it was described as the "lupus vulgaris pattern" by Ramam et al. [8] More recently, Williamson et al. proposed that the LGD pattern be considered as a new tuberculid variant [9].

The LGD pattern in lupus vulgaris comprises of tuberculoid granulomas forming a band-shaped infiltrate in the upper dermis and abutting the overlying hyperplastic epidermis. This is usually accompanied by small compact tuberculoid granuloma in deep dermis, which is mainly located in a peri-appendageal position [8]. It is important to differentiate lupus vulgaris from various closely related conditions such as tuberculosis verrucose cutis (TBVC), tuberculoid leprosy, deep fungal infections and secondary syphilis. All of these, at times, may demonstrate the LGD pattern in their histopathology. In addition, TBVC shows moderate to marked epidermal hyperplasia, hyperkeratosis and intra-epidermal neutrophilic micro-abscesses. Dermis may also show neutrophils in upper dermal infiltrate along with typical tuberculoid granulomas in deep dermis. Tuberculoid leprosy may show effacement of the rete ridge pattern of epidermis along with tuberculoid granulomas following the neurovascular bundles and involving the adnexal structures. Deep fungal infections characteristically show suppurative granulomatous dermatitis and organisms may be identified in PAS-stained sections. Secondary syphilis would require a close clinical correlation and serological investigations, as it can closely mimic almost any dermatological condition, clinically as well as histopathologically.

The LGD pattern was seen in our case, with a typical clinical appearance and response to a four-week therapeutic trial of anti-tuberculosis treatment, which clinched the diagnosis.

## 4. Conclusions

Lupus vulgaris is more common in developing countries, especially in the Asian continent. Thus, the LGD pattern is quite diagnostic of this condition in the right clinical settings. We described this case in order to make clinicians and histopathologists aware of this pattern in lupus vulgaris, so that the diagnosis is not missed in resource-poor settings and the patient receives timely treatment.

**Author Contributions:** Conceptualization, C.D., I.S.; methodology, C.D., I.S.; Software, C.D., I.S.; Validation, C.D., I.S.; Formal Analysis, C.D., I.S.; investigation, C.D., I.S.; resources, C.D., I.S.; data curation, C.D., I.S.; writing—original draft preparation, C.D.; Writing—Review and editing, C.D., I.S.; Visualization, C.D., I.S.; Supervision, C.D., I.S. All authors have read and agreed to the published version of the manuscript.

**Funding:** This research received no external funding.

**Institutional Review Board Statement:** Not applicable.

**Informed Consent Statement:** Informed consent was obtained from the patient to publish this paper.

**Data Availability Statement:** Not applicable.

**Conflicts of Interest:** The authors declare no conflict of interest.

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
