# Peer review of "“Lichenoid Granulomatous Pattern” in a Case of Lupus Vulgaris"

_dermatopathology, doi:10.3390/dermatopathology9020016_

Round 1
Reviewer 1 Report
Please change the word 'male' to 'boy'
The word vanished in the last paragraph may be replaced by regression
Author Response
|
Reviewer comments (1) |
Response by authors |
Site of changes made in the manuscript |
|
Please change the word 'male' to 'boy' |
Change done |
Line no. 27 |
|
The word vanished in the last paragraph may be replaced by regression |
Change done |
Line no. 48 |
|
|
|
|
|
|
|
|
Reviewer 2 Report
Interesting point brought out by the authors. Lichenoid granulomatous dermatitis is fairly commonly seen in lupus vulgarisms, though has not been well documented in the literature.
Was there any follicular plugging or necrosis seen in the biopsy?
Correct Raman et al in line no. 80 to Ramam et al.
Discuss how to differentiate LV from other close differentials like Tuberculosis verrucosa cutis, deep fungal infections, tuberculoid leprosy, secondary syphilis; some of these may show the lichenoid dermatitis pattern.
Since LV is more common in the developing countries esp Asian countries, this pattern is quite diagnostic of LV, this geographical aspect can be stressed upon in the text.
Author Response
|
Reviewer comments (2) |
Response by authors |
Site of changes made in the manuscript |
|
Interesting point brought out by the authors. Lichenoid granulomatous dermatitis is fairly commonly seen in lupus vulgarisms, though has not been well documented in the literature. |
Thank you |
|
|
Was there any follicular plugging or necrosis seen in the biopsy? |
No, we could not see any necrosis or follicular plugging in studied sections of our case |
|
|
Correct Raman et al in line no. 80 to Ramam et al. |
Correction done |
Line no. 80 |
|
Discuss how to differentiate LV from other close differentials like Tuberculosis verrucosa cutis, deep fungal infections, tuberculoid leprosy, secondary syphilis; some of these may show the lichenoid dermatitis pattern. |
Added
“It is important to differentiate lupus vulgaris from its close differentials like tuberculosis verrucose cutis (TBVC), tuberculoid leprosy, deep fungal infections and secondary syphilis. All of these at times may demonstrate LGD pattern on histopathology. TBVC in addition shows moderate to marked epidermal hyperplasia, hyperkeratosis and intra-epidermal neutrophilic microabcesses. Dermis may also show neutrophils in upper dermal infiltrate along with typical tuberculoid granulomas in deep dermis. Tuberculoid leprosy may show effacement of rete ridge pattern of epidermis along with tuberculoid granulomas following the neurovascular bundles and involving the adnexal structures. Deep fungal infections characteristically show suppurative granulomatous dermatitis and organisms may be identified in PAS stained sections. Secondary syphilis would require a close clinical correlation and serological investigations, as it can closely mimic almost any dermatological condition clinically as well as on histopathology. |
Line no. 85 to 97 |
|
Since LV is more common in the developing countries esp Asian countries, this pattern is quite diagnostic of LV, this geographical aspect can be stressed upon in the text. |
Added
“Lupus vulgaris is more common in developing countries, especially in the Asian continent. Thus, the LGD pattern is quite diagnostic of this condition in right clinical settings.” |
Line no. 102 and 103 |
Round 2
Reviewer 1 Report
Tuberculosis strictly speaking need not be given as a differential diagnosis since both conditions are of the sae etiology.
Author Response
Although, tuberculosis verrucose cutis shares the same etiology with lupus vulgaris, we wanted to highlight it as one of the differentials of LGD pattern and that there are subtle clues to differentiate it based on histopathology. We believe that it may not be important from the practical therapeutic viewpoint, but would be pertinent academically to enlist all possible differentials (for the sake of completeness).
Reviewer 2 Report
No additional changes.